# The Debatable Benefit of Gross-Total Resection of Brain Metastases in a Comprehensive Treatment Setting

**DOI:** 10.3390/cancers13061435

**Published:** 2021-03-21

**Authors:** Stephanie T. Jünger, Lenhard Pennig, Petra Schödel, Roland Goldbrunner, Lea Friker, Martin Kocher, Martin Proescholdt, Stefan Grau

**Affiliations:** 1Centre for Neurosurgery, Department of General Neurosurgery, Faculty of Medicine and University Hospital Cologne, University of Cologne, 50923 Cologne, Germany; stephanie.juenger@uk-koeln.de (S.T.J.); roland.goldbrunner@uk-koeln.de (R.G.); l.friker@gmail.com (L.F.); 2Centre for Integrated Oncology, Faculty of Medicine and University Hospital Cologne, University of Cologne, 50923 Cologne, Germany; 3Institute for Diagnostic and Interventional Radiology, Faculty of Medicine and University Hospital Cologne, University of Cologne, 50923 Cologne, Germany; lenhard.pening@uk-koeln.de; 4Department of Neurosurgery, University Hospital Regensburg, 93053 Regensburg, Germany; Petra.Schoedel@klinik.uni-Regensburg.de (P.S.); martin.proescholdt@gmail.com (M.P.); 5Wilhelm Sander Neuro-Oncology Unit and Department of Neurology, University of Regensburg, 93053 Regensburg, Germany; 6Centre for Neurosurgery, Department of Stereotactic and Functional Neurosurgery, Faculty of Medicine and University Hospital, University of Cologne, 50931 Cologne, Germany; martin.kocher@uk-koeln.de

**Keywords:** brain metastasis, extent of resection, comprehensive oncological treatment, systemic treatment

## Abstract

**Simple Summary:**

In this monocentric retrospective analysis, the extent of resection of singular/solitary brain metastases has no impact on local recurrence and overall survival rates in patients receiving multidisciplinary adjuvant treatment. Since systemic disease progression is the leading cause of death, and an uncontrolled systemic disease status, along with adjuvant treatment, present independent predictors of overall survival, a comprehensive, multidisciplinary treatment concept is essential for patients with brain metastases.

**Abstract:**

*Background and Purpose:* The value of gross-total surgical resection remains debatable in patients with brain metastases (BMs) as most patients succumb to systemic disease progression. In this study, we evaluated the impact of the extent of resection of singular/solitary BM on in-brain recurrence (iBR), focusing on local recurrence (LR) and overall survival (OS) in an interdisciplinary adjuvant treatment setting. *Patients and Methods:* In this monocentric retrospective analysis, we included patients receiving surgery of one BM and subsequent adjuvant treatment. A radiologist and a neurosurgeon determined in consensus the extent of resection based on magnetic resonance imaging. The OS was calculated using Kaplan–Meier estimates; prognostic factors for LR and OS were analysed by Log rank test and Cox proportional hazards. *Results:* We analyzed 197 patients. Gross-total resection was achieved in 123 (62.4%) patients. All patients were treated with adjuvant radiotherapy, and 130 (66.0%) received systemic treatment. Ninety-six (48.7%) patients showed iBR with an LR rate of 23.4%. LR was not significantly influenced by the extent of resection (*p* = 0.139) or any other parameter. The median OS after surgery was 18 (95%CI 12.5–23.5) months. In univariate analysis, the extent of resection did not influence OS (*p* = 0.6759), as opposed to adjuvant systemic treatment (*p* < 0.0001) and controlled systemic disease (*p* = 0.039). Systemic treatment and controlled disease status remained independent factors for OS (*p* < 0.0001 and *p* = 0.009, respectively). *Conclusions:* In this study, the extent of resection of BMs neither influenced the LR nor the OS of patients receiving interdisciplinary adjuvant treatment.

## 1. Introduction

With improvements in systemically-active anti-cancer therapies and due to the availability of more sophisticated diagnostic techniques, the number of patients diagnosed with brain metastases (BMs) is steadily increasing [1,2,3]. The surgical resection of BMs mainly serves three purposes: relieving eloquent brain areas to reduce neurological symptoms, achieving local control, and obtaining material for histopathological diagnosis [4].

Early studies demonstrated a significantly improved overall survival (OS) for patients with BMs receiving surgical resection plus adjuvant whole brain radiation therapy (WBRT) compared to WBRT alone [5,6]. However, these studies were conducted in a pre-radiosurgical era and before magnetic resonance imaging (MRI) was widely available. The implementation of postsurgical MRI enables an accurate assessment of BM resection [7,8,9]. Subsequently, a number of studies have demonstrated a better outcome in patients after complete resection of BMs [7,9,10,11,12]. In daily clinical practice, the major reasons for an incomplete resection of BMs are an eloquent localization of the tumor and the surgeon’s intention to avoid surgery-related neurological morbidity. Therefore, the postoperative presence of residual tumor is a non-random event, potentially influencing the outcome as a confounding factor [13,14]. However, with advanced imaging modalities [9] and histopathological analyses demonstrating BM invasion into the central nervous system (CNS) [15,16,17], the potential value of a radical resection remains debatable [18,19,20].

While a positive impact of a gross-total resection on local in-brain recurrence (iBR) appears probable, evidence in the literature regarding the positive effect upon OS is conflicting [9,18]. Considering the importance of local postsurgical treatment of BMs, even after complete surgical resection [21], we hypothesized that incomplete resection may not be associated with an inferior local control rate or decreased OS if adequate adjuvant treatment is applied.

The purpose of this study was to evaluate the impact of gross-total vs. sub-total resection of singular/solitary BM on time to iBR, focusing on local recurrence (LR) and its influence on the OS in patients receiving interdisciplinary adjuvant treatment.

## 2. Material and Methods

### 2.1. Patient Population and Data Collection

We retrospectively reviewed our internal database at our tertiary care medical centre for patients aged >18 years, who received surgery for a single BM and interdisciplinary adjuvant treatment from 2010 to 2019. Demographic and clinical parameters, including the systemic disease status of the patients, were retrieved from electronic and paper charts, which were documented using the Research Electronic Data Capture (REDCap, Vanderbilt University, Nashville, TN, USA) database. Information about the systemic disease status was acquired from radiological, oncological, and medical reports or tumor board protocols.

The time to diagnosis of BMs was defined as the period from primary tumor diagnosis to the date of surgery for BMs. The times to iBR and OS were calculated from the date of BM resection until intracerebral progression on MRI and death/last follow-up, respectively. Pre- and postoperative neurological status was classified according to the Medical Research Council-Neurological Performance Status Scale (MRC-NPS). The patients’ general condition was assessed using the Karnofsky performance scale (KPS). Patients were allocated to radiation therapy oncology group (RTOG) recursive partitioning analysis (RPA) classes [22,23].

Patients were excluded in cases of (1) previous treatment of BMs, (2) unavailable data of (radio-) oncological adjuvant treatment, (3) unknown extent of resection or local disease-control status, (4) death within the first month after surgery, (5) no postoperative radiotherapy, and (6) unavailable pre- or postoperative MRI.

### 2.2. Surgery

Surgery was indicated within an interdisciplinary institutional tumor board comprising board-certified neurosurgeons, neuro-oncologists, medical oncologists, neuro-radiologists, neuropathologists, and palliative care physicians.

Surgery was conducted under general anaesthesia by eight board-certified neurosurgeons. Optic neuronavigation for craniotomy planning was used on a regular basis for supratentorial tumors. Depending on eloquent tumor location, pre- and intraoperative brainmapping was conducted. If indicated, intraoperative ultrasound was used to detect residual tumor tissue.

### 2.3. Assessment of Tumor Burden, Eloquence, Extent of Tumor Resection, and Potential in-Brain Recurrence

To define the tumor burden, a radiologist and a neurosurgeon (L.P., S.G.) with four and 20 years of experience in neuro-oncologic imaging and MRI independently measured the largest diameter of BMs in three dimensions on pre-operative MRI of the head (on T1-weighted contrast-enhanced (T1 CE) sequence, including contrast- and non-contrast-enhancing tumor parts). The mean of both assessed volumes calculated from the three diameters was considered representative of tumor burden. Furthermore, in consensus, the experts assessed the BM location for potential eloquence. Eloquence was defined as previously published, comprising pre- and postcentral gyrus, speech, and visual cortex [24].

To define the extent of resection, in consensus, the experts evaluated postoperative MRI scans. As part of our institutional standard postoperative procedure, every patient received MRI of the head within 24 to 48 h after surgery using a standardized protocol comprising T1-/T2-weighted, T1 CE, and fluid-attenuated inversion recovery (FLAIR) sequences. Together, the experts conducted a review of the original radiology report and evaluated scans for the presence of residual tumor applying the following grading scale:0 = no residual tumor (GTR)1 = ≤5% residual tumor2 = 5–20% residual tumor3 = >20% residual tumor1–3 = sub-total resection (STR)

After discharge, patients received the same standardized protocol during regular follow-up visits. The interdisciplinary institutional tumor board determined the final diagnosis of a potential iBR. If differentiation between recurrence, radiation necrosis, or pseudoprogression under immune-/targeted therapy was inconclusive, F-18-Fluorethyltyrosin positron emission tomography (FET-PET) was performed. The pattern of iBR was classified based on the location of a new contrast-enhancing intracranial mass on T1 CE, which was considered suspicious for a BM:(1)local recurrence (LR): at resection cavity only(2)distant recurrence (DR): distant without contact to resection cavity(3)combined recurrence (CR): combination of LR and DR

### 2.4. Statistical Analysis

Statistical analysis was performed using SPSS software (release 25.0; IBM, Armonk, NY, USA). For descriptive statistics, continuous values are given in median or mean and range, ordinal and categorical variables are stated in numbers and percentages. Normal distribution of continuous variables was assessed using the Levene’s test. Ordinal and categorical variables were compared by the χ^2^ or Fisher’s exact test. The time to iBR and OS rates were estimated using the Kaplan–Meier method (log-rank test). Multivariate Cox regression was conducted using the pairwise inclusion method. *p*-values < 0.05 were considered statistically significant.

## 3. Results

### 3.1. Patient Demographics and Baseline Clinical Data

Between 2012 and 2019, 318 patients received surgery for a singular or solitary BM in our department. Ninety-two patients were excluded due to missing pre- or postoperative MRI, previous treatment of BMs, or missing data on adjuvant treatment. Furthermore, 29 patients were excluded as they did not survive the early postoperative phase of 30 days (in the majority of cases due to systemic disease progression and non-surgical, mostly cardio-embolic complications) and, therefore, did not receive adjuvant local treatment. Consequently, 197 patients were included in this study.

The median age at the time of diagnosis of BMs was 62 (19–87) years. Gender distribution was almost equal with 88 (44.7%) male and 109 (55.3%) female patients. Primary tumors (PT) comprised non-small-cell lung cancer in 92 (46.7%), malignant melanoma in 20 (10.2%), breast cancer in 27 (13.7%), tumors of the gastro-intestinal tract in 33 (16.8%), and others/rare entities in 17 (8.6%) patients. Cancer of unknown primary (CUP)-syndrome was present in seven (3.6%) patients (Table 1).

The median time between diagnosis of PT and BM was 14.1 months (range: 11.5 to 312 months). The systemic disease at the time of BM diagnosis was controlled in 63 (32.0%) patients. In 74 (37.6%) patients, BMs occurred in an eloquent location (see Table 1 for locations).

The mean preoperative tumor burden was 41.6 cm^3^ (range: 1.0–240.1 cm^3^). Before surgery, the median KPS was 80 (range 40–100), and the neurological status was assessed by the MRC-NPS (Table 2). Pre-operative patient allocation according to RTOG RPA classes resulted in 55 (27.9%) patients in class I, 96 (48.7%) in class II, and 46 (23.4%) in class III (Table 2).

### 3.2. Treatment-Related Parameters

The extent of resection was rated as gross total in 123 (62.4%) patients and sub-total in 74 (37.6%), comprising remnants of ≤5% in 62, 5–20% in 10, and >20% in two patients. No patient received additional surgery for residual tumor. The preoperative tumor volume did not correlate with the extent of resection (rho 0.06, *p* = 0.401).

The gross-total resection rate differed between the tumor entities. Complete resection was achieved more frequently in the cerebellum (71.2%) than in the supratentorial lobes; however, the individual differences regarding individual tumor entities and location did not reach significance (*p* = 0.626). Gross-total resection rates were significantly lower for BMs at an eloquent location (*p* = 0.004). All other parameters (age, clinical status, time between initial diagnosis of disease and occurrence of BM, adjuvant treatment modality, primary tumor, neurological symptoms, and tumor volume) were not significantly different between patients undergoing gross-total resection vs. sub-total resection.

The MRC-NPS significantly improved after surgery (*p* < 0.001). Consequently, a significant improvement in the KPS was observed (90 (range: 30–100); *p* < 0.001). The extent of resection did not influence the improvement in the KPS or MRC-NPS after surgery (*p* = 0.419 and 0.681, respectively). Recursive partitioning analysis (RPA) class allocation was significantly improved by surgery with 69 (35.0%) patients in class I, 121 (61.4%) in class II, and seven (3.6%) in class III (*p* < 0.001) (Table 2).

Nine (4.0%) patients suffered from surgery-related complications. These comprised postoperative haemorrhage (*n* = 1), pulmonary artery embolism (*n* = 2), cerebrospinal fluid fistula (*n* = 1), acute ischemic stroke (*n* = 1), and wound infection (*n* = 4).

Postoperative cerebral radiotherapy was applied in all patients, comprising local fractionated radiotherapy (fRT) in 142 (72.1%) [25,26,27,28], neo-/adjuvant stereotactic radiosurgery (sRS) in six (3.0%), and WBRT in 49 (24.9%) patients. Radiotherapy was performed in-house as well as at a range of collaborating institutes. In most cases, target definition for fRT was based on a planning CT registered to the early postoperative MRI, where the resection cavity and any residual contrast-enhancing regions in the early postoperative MRI were used to define the gross tumor volume (GTV). For the clinical target volume (CTV), circular margins of 1.0–1.5 cm were typically added such that the dura was included when the cavity reached the surface of the brain. In most cases, 3D-conformal irradiation was applied within 4–6 weeks after surgery using linear accelerators (6–15 MV photons) equipped with multileaf collimators and using multiple static beams or volumetric arc techniques. Most patients received 36.0–45.0 Gy in 3.0 Gy, 40.0–50.4 Gy in 1.8–2.0 Gy or 41.8 Gy in 3.8 Gy fractions. WBRT was mostly carried out in the collaborating institutes using standard techniques and doses of 30–36 Gy. For sRS, single doses of 20 Gy (neoadjuvant) or 10 Gy (adjuvant) were applied.

### 3.3. Cerebral Disease Control and Overall Survival

Regular follow-up imaging after initial postoperative MRI was available for 185 patients. The median radiological follow-up was nine (3–104) months. During the follow-up period, 97 (49.2%) patients suffered from iBR. The recurrence pattern was LR in 23 (23.7%) patients, DR in 63 (64.9 %), and CR in eleven (11.3 %).

The actuarial rates for freedom from iBR after 6, 12, and 18 months were 75, 52, and 42%, respectively. Postoperative WBRT reduced the iBR but without reaching statistical significance (*p* = 0.055).

The rates of local in-brain recurrence differed among the individual PTs with breast cancer, leading to local recurrences more frequently than other cancers but without reaching significance (*p* = 0.713). The same applied to the supra-/infratentorial (*p* = 0.372) or eloquent (*p* = 0.769) localization of BMs.

LR control rates after 6, 12, and 18 months were 89%, 80%, and 68%. The incidence of LR was not significantly influenced by the extent of resection (*p* = 0.139, Figure 1) nor the modality of applied radiotherapy (*p* = 0.154) or any other parameter assessed (Table 3).

By the time of analysis, 127 (64.5%) patients had died; the cause of death could be reliably identified in 121 patients, as systemic tumor progression in 78, neurologic in 26, and non-tumor related in 17 patients. No correlation was observed between the extent of resection and the cause of death (sub-total vs. gross-total resection *p* = 0.277 or residual tumor burden *p* = 0.267). Furthermore, neither PT nor BM location showed a significant impact on OS (*p* = 0.355 and *p* = 0.624, respectively); an eloquent location did not influence OS either (*p* = 0.260).

The median clinical follow up was 12 (3–119) months with a median survival after surgery of 12 (95%CI 8.6–154) months.

In univariate analysis, the extent of resection did not influence OS (log-rank test, *p* = 0.759, Figure 2); in other words, a greater extent of resection was not associated with prolonged OS, while postoperative systemic treatment (log-rank test, *p* < 0.0001) and controlled systemic disease (log-rank test, *p* = 0.039) were significant. In regression analysis, only systemic treatment (HR 0.45 95%CI 0.31–0.65; *p* < 0.0001) and controlled disease status (HR 0.59 95%CI 0.40–0.88; *p* = 0.009) remained independent factors for survival (Table 3).

## 4. Discussion

The importance of surgery for patients with single/solitary, large, and symptomatic BM is well established, but the relevance of the extent of resection for cerebral control and the patient’s further OS is contentious [7,9,10,11,12,18,29,30,31,32,33,34]. We, therefore, aimed to clarify its role regarding iBR, focusing on LR in particular and OS in a well-defined and large cohort that underwent standardized evaluation of the extent of resection by early postoperative MRI and consistent adjuvant local and systemic treatment.

The issue of early MRI control after tumor resection has been raised by studies on glioblastoma, where complete resection of contrast-enhancing tumor tissue has shown a progression-free survival benefit [35]. Transferring this idea to patients suffering from BM is not self-evident since, in cancer patients, the brain is not the only location but rather one potential manifestation site of a systemically spreading disease. Therefore, cerebral treatment may not entirely define the patient’s oncological course. Consequently, assuming that controlled local cerebral disease per se improves survival is not logical.

The gross-total resection rate of 62.4% reported in this study is within the scope of previously reported values of 61.5 or 57.0% from studies by Kamp et al. or Olesrud et al. [7,9]. Lee et al. reported gross-total resection rates of 75.8%; however, covering a time span of 17 years, post-operative imaging modalities might not be comparable [11]. Of note, the sub-total resection rate in this study of 38.6% exceeds those in the above-mentioned studies by Kamp and Olesrud (20 and 22%, respectively) [7,9]. An explanation may be that the considerable number of patients in those two studies precluded determining the gross-total resection on MRI [7,9] or that the extent of resection was not assessed by radiologists.

Along with the gross-total resection rates, an iBR rate of 48.7% with DR occurring more frequently than LR is comparable to previous studies regarding frequency and distribution [7,36]. However, opposed to previous studies, we could detect neither a significant correlation between gross-total resection and iBR (in terms of LR) [7] nor with OS [5,6,9,37].

The major reason for our differing findings may be seen in the different study populations and the postsurgical treatment structure: whereas in our cohort, only 32% of all patients presented with controlled systemic disease, studies showing benefits of complete resection mostly included patients with controlled systemic disease (68.5–79.0%) [10,11,12]. Since, in these studies, the CNS was the only site of tumor spread in most patients, oncological control of this site may lead to improved survival. In these studies, cause of death was not reported in detail; however, a retrospective study by Lee et al. reports systemic disease progression as responsible for 75% of deaths compared to 25% due to fatal cerebral progression [11].

Despite administering postoperative radiation therapy to all patients of this present cohort, the incidence of overall iBR was in line with recently published patient cohorts [7,18]; however, only a minor fraction (24%) occurred at the site of the operated BM. We are well aware that the most frequently applied fRT may not represent the standard adjuvant radiation modality [21,38], but previous data indicate promising results [36]; therefore, our results regarding LR may be representative. Nonetheless, with a reasonable number of patients treated with WBRT, the incidence of iBR may not be reliably correlated with the extent of resection, since WBRT compared to fRT and radiosurgery does not show restriction to the resection site.

Regarding systemic postsurgical treatment, the majority of previously published studies did not provide information on this crucial aspect. In our cohort, 66.0% of all patients received systemic treatment, which may have significantly influenced the disease course, particularly in patients with sub-total resection, since reasonable intracranial response rates upon targeted and immune treatments have been reported [39,40]. Our differing results must be interpreted in the specific cohort constellation comprising only patients with a reasonable clinical status or a fair chance of improvement after surgery, single or solitary BM, postoperative radiation therapy, and a high percentage of subsequent systemic treatment. In this context, this study implies a relevant selection bias by including only patients with a fair clinical status (with the exception of a few acute deteriorations by the BM) and a potential further treatment concept.

In this study, for patients in a good clinical condition, resection of symptomatic BM leads to a significant improvement in general and neurological status. This, in turn, may facilitate further treatment, since the application of systemic therapy in particular is frequently restricted to patients with a good functional status [41]. The fact that a KPS ≥ 70 did not show an impact on survival in this present study may be due to the positive selection bias of this present cohort since patients who did not survive the early postoperative phase and who consequently did not receive any adjuvant radiation treatment were excluded. With respect to the main question of this study, the important issue is that we could demonstrate equivalent neurological and functional improvement rates regardless of the extent of resection, and that residual tumor did not lead to repeated surgery in one single case.

The surgical treatment paradigm in oncological neurosurgery is a maximal tumor resection with minimal morbidity [4]. In clinical practice, the most prevalent reason for a sub-total resection is the attempt to avoid surgery-induced neurological morbidity during the resection of an eloquently located lesion or an association between tumor and brain vasculature [42]. Our results indicate that reduction in intracranial pressure, acquisition of tissue to establish histological and/or molecular diagnoses, and reduction in neurological symptoms can frequently be achieved with, even if not intended, sub-total resection. In this context, there is no justification for radical surgery at the cost of functional deterioration.

Indeed, the point above gains more weight since our LR rates are within the scope of previously reported rates in supra-marginal resected BM (22% vs. 14.9–29.1%) [18,20]. Although, Yoo et al. confirmed supra-marginal resection by histopathological analysis, their local recurrence rates were not superior to those presented here. However, in their cohort, not all patients received adjuvant treatment. Additionally, it should be mentioned that they were able to demonstrate a reasonable advantage in their cohort comparing supra-marginal resection to “normal” resection [18]. Nevertheless, in the case of secured adjuvant radiotherapy, supra-marginal resection may not be needed.

In clinical neurosurgical practice, achievement of long-lasting cerebral disease control is frequently pursued by neurosurgeons assuming that cerebral control has a direct impact on OS. However, to date, no larger trial has demonstrated a correlation between local cerebral control and a patient’s survival [21,43], a fact reflecting the complex condition of the patient, who suffers from not only a local cerebral problem, but also a systemic one. One possible explanation may be that systemic disease progression, in line with the results from Yoo et al. [18], represents the leading cause of death in this present study.

This study does not aim to make a case against surgery for BM but intends to underline the importance of clinical improvement over radicality. Our results also suggest that (unintended) sub-total resection, when considering patients integrated into a highly functional multidisciplinary treatment matrix, may convey sufficient local tumor control. Especially regarding surgery in eloquent brain areas, the extent of resection needs to be fine-tuned between providing adequate tumor control and improving/preserving function, while at the same time avoiding surgery-induced neurological deficits.

We are aware that the follow-up period of some patients is rather short; however, we have a reasonable number of long-term survivors, possibly due to comprehensive adjuvant treatment. Furthermore, even though a reasonable number of patients received immuno- and/or targeted therapies, explicit deductions about the effects of these therapies in combination with radiotherapy could not be made, because (i) groups of PT were heterogeneous and often very small; (ii) the molecular status was not known for all PT and BM, and (iii) adjuvant radiation differed, making a standardized analysis of this aspect very difficult. Nevertheless, due to its importance, it definitely deserves further investigation in the future.

## 5. Conclusions

In conclusion, the risk of postoperative impairment should not be traded against a gross-total resection of BMs since the extent of resection has no impact on neurological status, LR, or OS in the setting of multimodal adjuvant treatment.

## Figures and Tables

**Figure 1 cancers-13-01435-f001:**
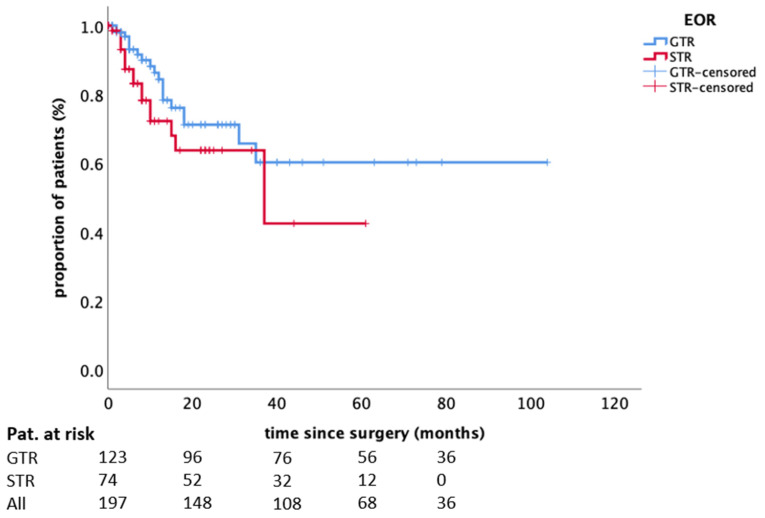
The influence of the extent of resection on local recurrence. Kaplan–Meier plot (*n* = 197; incomplete/complete: 74/123; *p* = 0.1395). Abbreviations: EOR—extent of resection, GTR—gross-total resection, STR—sub-total resection.

**Figure 2 cancers-13-01435-f002:**
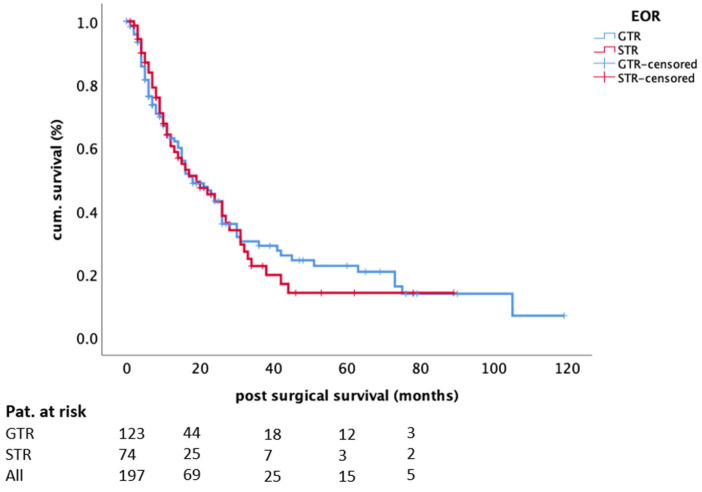
The influence of the extent of resection on OS. Kaplan–Meier plot (*n* = 197; incomplete/complete: 74/123; *p* = 0.759). Abbreviations: EOR—extent of resection, GTR—gross-total resection, STR—sub-total resection.

**Table 1 cancers-13-01435-t001:** Patient demographics, baseline characteristics, and location of tumors.

Parameter	Value
**Age [median; (range)]**	62 (19–87)
**Gender [n; (%)]**	
Male	88 (44.7)
Female	109 (55.3)
**Primary [n; (%)]**	
Lung (NSCLC)	92 (46.7)
Melanoma	20 (10.2)
Breast	27 (13.7)
Gastrointestinal	33 (16.8)
Others	17 (8.6)
Cancer of unknown primary	7 (3.6)
**Controlled systemic disease [n; (%)]**	63 (32.0)
**Tumor location [n; (%)]**	
Frontal	65 (33.0)
Temporal	29 (14.7)
Parietal	21 (10.7)
Temporo-parietal	3 (1.5)
Occipital	12 (6.1)
Cerebellar	66 (33.5)
Basal ganglia	1 (0.5)
**Eloquent location [n; (%)]**	74 (37.6)
**Neurological deficits [%]**	
Seizures	15.7
Aphasia	8.1
Hemiparesis	29.9
Visual field defects	6.6
Signs of elevated intracranial pressure	33
**Adjuvant Treatment [n; (%)]**	130 (66.0)
Systemic medical treatment	75 (38.1)
Molecular treatment	
**Postoperative Radiotherapy**	
Whole brain radiotherapy	47 (24.9)
Partial/focal radiotherapy	142 (72.1)
Stereotactic radiosurgery	6 (3.0)

**Table 2 cancers-13-01435-t002:** Clinical scores and prognostic group allocation. Statistic changes in group allocation, calculated by Wilcoxon rank-sum test.

Scale	Before Surgery	After Surgery	*p*-Value
**MRC-NPS [n; (%)]**			**<0.001**
1	25 (12.7)	110 (55.8)	
2	97 (49.2)	68 (34.5)	
3	31 (15.7)	14 (7.1)	
4	44 (22.3)	4 (2.0)	
5		1 (0.5)	
**KPS [median; range]**	80 (40–100)	90 (30–100)	**<0.001**
**RPA group [n; (%)]**			**<0.001**
1	55 (27.9)	69 (35.0)	
2	96 (48.7)	121 (61.4)	
3	46 (23.4)	7 (3.6)	

Abbreviations: KPS—Karnofsky performance scale, MRC-NPS—medical research council neurological performance scale, RTOG—radiation therapy oncology group, RPA—recursive partitioning analysis.

**Table 3 cancers-13-01435-t003:** Parameters predicting local recurrence and overall survival. Bold indicates statistical significance, calculated by log-rank test and cox regression.

Parameter	Local Recurrence	Overall Survival	
	Log Rank Test (*p*-Value)	Log Rank Test (*p*-Value)	Cox Regression (HR 95%CI; *p*-Value)
**Age < 65 years**	0.780	0.251	
**Gender**	0.144	0.100	
**Controlled systemic disease**	0.698	0.039	**0.59 0.40–0.88 0.009**
**Extent of resection (GTR vs. STR)**	0.139	0.759	
**Adjuvant systemic treatment**	0.313	<0.0001	**0.45 0.31–0.65 <0.0001**
**Adjuvant radiotherapy modality (WBRT vs. fRT vs. SRS)**	0.154		
**In-brain recurrence**		0.114	
**Local recurrence**		0.591	

Abbreviations: GTR—gross-total resection, STR—sub-total resection, HR—hazard ratio, CI—confidence interval.

## Data Availability

The datasets used and/or analyzed during the current study are available from the corresponding author on reasonable request.

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
