# Peer review of "The Debatable Benefit of Gross-Total Resection of Brain Metastases in a Comprehensive Treatment Setting"

_cancers, 2021, doi:10.3390/cancers13061435_

Round 1
Reviewer 1 Report
Review:
The debatable benefit of gross-total resection of brain metastases in a comprehensive treatment setting
The authors appear to make a case against surgery for patients with a single brain metastasis since there is no LR or OS benefit. As was stated in the introduction, there are at least 3 reasons for surgery for these patients and it was shown that there were neurological improvements after surgery; there should be recognition of that in the conclusion. In addition, the authors really sum up the limitation of their study in lines 257-258: only 32% of the study patients had controlled systemic disease. This suggests that the main reason for surgery was neurological relief and perhaps local control, not OS. The outcome therefore isn’t surprising: a local treatment cannot reverse active systemic disease.
Figure 2 has a lot of censored events, patient count goes to zero but many are still alive. For example, in the GTR arm, the data doesn't even show a median survival yet (> 50% of patients are still alive). Critical survival information appears to be lacking.
There is next to no information about the post-op radiotherapy. Please add dose and fractionation information.
At line 219, you mention that KPS had no effect on OS. Were there enough patients with low KPS to properly assess this? I would expect that patients with low KPS often would not be eligible for surgery, as also reflected in your high median KPS score, in which case you would not be able to determine an effect of KPS on anything. You hint at this in line 284-286. It might be better to acknowledge fully that your patient selection was too biased to find effects of KPS.
L 24 better to use a colon rather than a comma
L 35 and Table 3 extent (noun, not the verb); as used in l 231!
L 44 This would read better if you replace `previous` with `early`.
L17 vs l56 spelling of debatable/debateable
L 68 the usual phrase is ‘single institution study’
L 81 what was the time lag between brain met diagnosis and surgery – how does this affect the analysis?
L 85 using the Karnofsky…. (not ‘by’)
L 105 representative of tumour burden
L 111 comprised of
L 120 what was the follow-up interval? Relates to exclusion 4 at l 90?
L 121 location rather than localization
Table 1. separate Adjuvant Treatment section
L 152 status of systemic disease was checked in 32% of patients. Do you mean ‘controlled’ instead of ‘checked’, as in Table 1? Using ‘checked’ sounds like you didn’t bother to look up the information for the other 68% of patients.
L 155 a very large range in tumour size. This warrants more discussion. Any effect on outcome?
L 167 ‘of 5-20’ can simply be ‘5-20’
L 171 ‘however, none reached significance’ – it’s not clear what comparison you have made.
L 196-202 at 6 months, for example, 58% of patients did not have new mets (new or regrowth); also, 92% did not have a local recurrence; hence there must be a large portion who had distal mets. I think this could be stated more clearly. Also, the information at line 202 should come right after the information in line 196.
L 206 Table 3 label: log-rank test (not text).
Table 3, the assignment of the 3rd column is not clear.
L 209 remove first comma in this line
L 209-210 neurologic
L 240 comma after `site`.
L 242 are you talking about controlled local disease in the brain here? That could be clarified more.
L 245-247 not clear which study took 17 years. According to your references, it would be Ref 11 (Lee), but this reference isn’t sited. Altogether not a very clear paragraph, too much use of `this study`, `present study`, without being clear which study is being referred to. Please rewrite this paragraph.
L 248-250 You appear to be saying that there is a difference in methodology. E.g. the sub-total resection rate in the other studies is based on evaluation at time of surgery, rather than post-surgery MRI. If this is the case, please clarify in your discussion.
L 250 what does ‘it’ refer to? Do you mean ‘or’ instead of ‘and’?
L 256-264. This paragraph sites a very significant difference in populations and points to a bad selection of patients for the study under review.
L 257 either add a comma after whereas, or remove the one after cohort.
L 258 with controlled systemic disease (take out ‘a’).
L 265 paragraph. This sounds like an excuse for not having done WBRT. It may be helpful to elaborate a bit on why you chose to use fRT and what that consisted of. 49 (for WBRT) is a low number; I don’t understand the logic of the last sentence of this paragraph. There seems to be a lot of reasoning going on in this paragraphs, but the thought process isn’t made clear.
L 271 treated with
L 275 disease course or course of disease
L 277-278 please specify how your results differ.
L 279 ‘only good clinical status’? You have KPS values as low as 30 and only 32% of your patients have controlled systemic disease. How does this represent good clinical status?
L 273 paragraph. It is not clear what information the authors are trying to convey.
L 283 no comma after particular
L 285 reference to `above-mentioned` not clear.
L 302-303 weird sentence, please clarify.
L 305 ‘also’ is in the wrong place. Our results also suggest… or perhaps you mean ….may also convey (on line 306).
L 307-309 another weird sentence, please rewrite.
L 310 very brief conclusion.
L 337 magnet resonance?
Author Response
Dear Reviewer,
thank you your comments, which were excellent, and we greatly appreciated your insights and critical suggestions. We have made major changes to the manuscript and believe we have addressed all the concerns.
On behalf of all the co-authors,
Stefan Grau M.D.
Reviewer 1
The authors appear to make a case against surgery for patients with a single brain metastasis since there is no LR or OS benefit. As was stated in the introduction, there are at least 3 reasons for surgery for these patients and it was shown that there were neurological improvements after surgery; there should be recognition of that in the conclusion. In addition, the authors really sum up the limitation of their study in lines 257-258: only 32% of the study patients had controlled systemic disease. This suggests that the main reason for surgery was neurological relief and perhaps local control, not OS. The outcome therefore isn’t surprising: a local treatment cannot reverse active systemic disease.
We regret this perception, since we definitely do not make a case against surgery, but rather question the paradigm of radicality, which was established for glioma and is frequently transferred to metastasectomy. In these patients the issue of symptom improvement goes beyond a palliative intention but influences access to further treatment and thus survival.
Although the reviewer and the authors absolutely share the perception of a complex disease constellation (“The outcome therefore isn’t surprising…”) the widespread attitude among many neurosurgeons still is very different. We therefore strengthened this aspect in the discussion. (l. 365-371; 372-373; 382-388)
Figure 2 has a lot of censored events, patient count goes to zero but many are still alive. For example, in the GTR arm, the data doesn't even show a median survival yet (> 50% of patients are still alive). Critical survival information appears to be lacking.
There is obviously a misinterpretation based on the wrong sequence of plots in the draft.
The mortality was 59% in the draft. The plot that the reviewer is referring to is the plot showing radiological control. We apologize for this mistake. (Figure 1 and 2)
There is next to no information about the post-op radiotherapy. Please add dose and fractionation information.
We added further data and included an experienced radio-oncologist (M. Kocher) in the manuscript. Of note, the RT concepts at our institution may vary from most other centres as we aim at strictly limiting irradiation dosage and therefore pursue focal radiotherapy. (l. 213-227)
At line 219, you mention that KPS had no effect on OS. Were there enough patients with low KPS to properly assess this? I would expect that patients with low KPS often would not be eligible for surgery, as also reflected in your high median KPS score, in which case you would not be able to determine an effect of KPS on anything. You hint at this in line 284-286. It might be better to acknowledge fully that your patient selection was too biased to find effects of KPS.
This absolutely correct! Obviously we have not made that clear enough in the manuscript! Also the exclusion of patients with early death contributed to this biased selection. As the main question behind this study was the impact of the EOR in a comprehensive treatment setting including postoperative radiotherapy these selection criteria are justified. We strengthened and clarified this in the revised version. (l. 327-333; 337-342)
L 24 better to use a colon rather than a comma
L 35 and Table 3 extent (noun, not the verb); as used in l 231!
L 44 This would read better if you replace `previous` with `early`.
L17 vs l56 spelling of debatable/debateable
L 68 the usual phrase is ‘single institution study’
L 85 using the Karnofsky…. (not ‘by’)
L 105 representative of tumour burden
We have changed all these aspects according to the suggestion.
L 111 comprised of
According to our language editor it is either “composed of” or “comprised”
L 121 location rather than localization
Table 1. separate Adjuvant Treatment section
L 167 ‘of 5-20’ can simply be ‘5-20’
L 171 ‘however, none reached significance’ – it’s not clear what comparison you have made.
L 206 Table 3 label: log-rank test (not text).
L 209 remove first comma in this line
L 209-210 neurologic
L 240 comma after `site`.
L 242 are you talking about controlled local disease in the brain here? That could be clarified more.
L 257 either add a comma after whereas, or remove the one after cohort.
L 258 with controlled systemic disease (take out ‘a’).
L 271 treated with
L 275 disease course or course of disease
L 277-278 please specify how your results differ.
L 283 no comma after particular
L 285 reference to `above-mentioned` not clear.
L 302-303 weird sentence, please clarify.
L 310 very brief conclusion.
L 337 magnet resonance?
L 305 ‘also’ is in the wrong place. Our results also suggest… or perhaps you mean ….may also convey (on line 306).
L 307-309 another weird sentence, please rewrite.
We have changed all these aspects according to the suggestion.
L 81 what was the time lag between brain met diagnosis and surgery – how does this affect the analysis?
This question probably aims at a potential previous – medical – treatment before surgery. We had no such patients. The interval between first diagnosis and surgery did not exceed two weeks, therefore no statistical analysis was possible.
L 120 what was the follow-up interval? Relates to exclusion 4 at l 90?
Absolutely correct. We updated this.
L 152 status of systemic disease was checked in 32% of patients. Do you mean ‘controlled’ instead of ‘checked’, as in Table 1? Using ‘checked’ sounds like you didn’t bother to look up the information for the other 68% of patients.
We apologize for this misunderstanding. The problem was the word “status”, which cannot be controlled, so in fact it was the systemic disease that was controlled. We changed this.
L 155 a very large range in tumour size. This warrants more discussion. Any effect on outcome?
There was no correlation between the initial tumour size nor the remnant size and outcome. We added this to the manuscript. (l. 193-194; 260-262)
L 196-202 at 6 months, for example, 58% of patients did not have new mets (new or regrowth); also, 92% did not have a local recurrence; hence there must be a large portion who had distal mets. I think this could be stated more clearly. Also, the information at line 202 should come right after the information in line 196.
The focus of this study was clearly set on the recurrence at the initial treatment site. As we had applied local irradiation in the majority of patients, tumour relapse at distant brain regions was mentioned, however, it does not contribute information to prove our hypothesis. We added further aspects to the manuscript. (l. 354-357)
Table 3, the assignment of the 3rd column is not clear.
We clarified this in the table. (Table 3)
L 245-247 not clear which study took 17 years. According to your references, it would be Ref 11 (Lee), but this reference isn’t cited. Altogether not a very clear paragraph, too much use of `this study`, `present study`, without being clear which study is being referred to. Please rewrite this paragraph.
We changed this. (l. 293.299)
L 248-250 You appear to be saying that there is a difference in methodology. E.g. the sub-total resection rate in the other studies is based on evaluation at time of surgery, rather than post-surgery MRI. If this is the case, please clarify in your discussion.
The weakness in many studies lies in the heterogeneous assessment of residual tumours (surgeon´s assessment, CT, MRI of varying quality). Therefore we can fill this gap due to standardized post-op imaging at an early time interval.
L 250 what does ‘it’ refer to? Do you mean ‘or’ instead of ‘and’?
We clarified this.
L 256-264. This paragraph cites a very significant difference in populations and points to a bad selection of patients for the study under review.
We do not agree with the perception of a bad selection. Apart from a selection bias (fair clinical status) we think that our population represents a real-life condition more than most other studies. In detail, some of the mentioned studies focussed on solitary (and not single) brain metastases, which represent a rather scarce condition in NSCLC patients.
Changed according to the suggestion.
L 265 paragraph. This sounds like an excuse for not having done WBRT. It may be helpful to elaborate a bit on why you chose to use fRT and what that consisted of. 49 (for WBRT) is a low number; I don’t understand the logic of the last sentence of this paragraph. There seems to be a lot of reasoning going on in this paragraphs, but the thought process isn’t made clear.
In the light of reported side effects of WBRT there is a general tendency towards local treatments over the last decade. In this context also partial or focal radiotherapy after surgery is established (e.g. Ayas et al. 2018) as a postoperative treatment option. We are aware that this does not meet standard treatment in many other centres. We have added substantial information to the manuscript about radiotherapy. (l. 213-227)
L 279 ‘only good clinical status’? You have KPS values as low as 30 and only 32% of your patients have controlled systemic disease. How does this represent good clinical status?
Absolutely correct, we clarified this issue further, since most patients with a poor KPS presented with large BM and had a reasonable chance of improvement of their general condition due to reduction of neurological symptoms. Good clinical status, should indicate that patients are ‘good’ enough to undergo further treatment. (l. 327-333)
L 273 paragraph. It is not clear what information the authors are trying to convey.
We do not understand this comment.

Reviewer 2 Report
The authors presented their interesting point of view on GTR in brain metastases coroborated by their large case-series.
According to their results, GTR does not provide an additional benefit on OS compared to STR. Unfortunately, GTR was retrospectively defined on post-operative MRI and no attempt were made to obtain a resection beyond metastasis boundaries at the time of the surgery.
I would suggest authors to briefly discuss their results compared to the effects of supra-total resection of brain metastases (SpTR), which is being recently debatet in literature.
Author Response
Dear Reviewer,
thank you your comments, which were excellent, and we greatly appreciated your insights and critical suggestions. We have made major changes to the manuscript and believe we have addressed all the concerns.
On behalf of all the co-authors,
Stefan Grau M.D.
Reviewer 2
The authors presented their interesting point of view on GTR in brain metastases corroborated by their large case-series.
According to their results, GTR does not provide an additional benefit on OS compared to STR. Unfortunately, GTR was retrospectively defined on post-operative MRI and no attempt were made to obtain a resection beyond metastasis boundaries at the time of the surgery.
I would suggest authors to briefly discuss their results compared to the effects of supra-total resection of brain metastases (SpTR), which is being recently debatet in literature.
Thank you for this comment; we included a paragraph addressing this issue. (l. 354-362)
Reviewer 3 Report
I am a neurosurgeon and translational clinical researcher with PhD and grants in brain metastases imaging and biology. As such I really enjoyed this interesting and original article which challenged the dogma that complete resection of brain mets is de facto going to extend survival. There is a big debate to be had about what our role is as neurosurgeons in this disease - does local control actually matter and actually, systemic disease is key. The article does have some presentation issues and needs more details to be really reliable however, see below suggestions and queries.
1. Were these all truly solitary BM? Or one dominant BM with smaller lesions that were not being treated surgically.
2. Why not calculate volume using software and extent of resection as %? Is it too late to go back and do this, it would be considerably more convincing than your kind of made up scale of GTR/STR which is not validated, not really used by anybody. Also, you could then analyze extent of resection as a continuous variable rather than just a category which will lose some detail.
3. Did you account for leptomeningeal recurrence or progression?
4. Were you using RANO or similar criteria to define progression?
5. You seem to indicate 29 patients died within 30d and could not be included? this is very high isn't it, like 10%? please clarify. Is it due to high rate of complications or poor performance status?
6. Can you rate eloquence in some way e.g. Sawaya grading system, left hemisphere vs right hemisphere, clarify what you mean by this - is it just speech, or motor areas or what
7. Clarify lung subtype - assume NSCLC.
8. RPA group is very crude, even the GPA now is primary disease specific now, any attempt to use other scores like this?
9. The major factor here is the surgical technique - are you leaving residual because it is unsafe or because you are having variation in the surgery. SO much more detail please on the surgery - how many surgeons in this series? What standard techniques e.g. image guidance? ultrasound? What novel techniques like 5-ALA. Any adjuncts like motor mapping, awake or iMRI?
10. It may be that your reasons for having great outcomes despite subtotal resection is just that you are doing something different with radiation - so you MUST clarify local the doses, machine (LINAC?), type of radiation - how many fractions, what do you do about dura, the margin, the cavity, when do you irradiate after surgery, what scans do you use to plan etc etc
11. What is the role of immunotherapy in this series
12. Table 3 is poorly displayed - do you mean this is univariate analysis of each factor or multivariate? Please state the time to LR and death in months for each level of the factors assessed e.g. >65y and <65 y old, GTR and STR then then univariate test statistic, then the multivariate if it was taken forward to multivariate analysis.
13. Only 60% of patients have died?? Line 208. This introduces significant lead time bias. Why are you reporting this series so early? Perhaps you should not include the more recent patients e.g. last couple of years. This could be a major issue.
14. Line 217 - please be much less vague and much more specific in your language, X increases Y, not "influenced" e.g. greater extent of resection was not associated with prolonged overall survival in this series (test statistic = X, p value = Y).
Author Response
Dear Reviewer,
thank you your comments, which were excellent, and we greatly appreciated your insights and critical suggestions. We have made major changes to the manuscript and believe we have addressed all the concerns.
On behalf of all the co-authors,
Stefan Grau M.D.
Reviewer 3
I am a neurosurgeon and translational clinical researcher with PhD and grants in brain metastases imaging and biology. As such I really enjoyed this interesting and original article which challenged the dogma that complete resection of brain mets is de facto going to extend survival. There is a big debate to be had about what our role is as neurosurgeons in this disease - does local control actually matter and actually, systemic disease is key. The article does have some presentation issues and needs more details to be really reliable however, see below suggestions and queries.
- Were these all truly solitary BM? Or one dominant BM with smaller lesions that were not being treated surgically.
All the patients had a single BM, solitary (as defined by exclusively BM and no other mets) BM were fewer (as reflected by the controlled disease status). All Patient had a pre-op MRI and we had excluded patients with pre-op CT imaging only. (l. 88; 103-106)
- Why not calculate volume using software and extent of resection as %? Is it too late to go back and do this, it would be considerably more convincing than your kind of made up scale of GTR/STR which is not validated, not really used by anybody. Also, you could then analyze extent of resection as a continuous variable rather than just a category which will lose some detail.
We appreciate this comment. Indeed, we intended to compute a volumetric analysis. However, as the majority of patients received their imaging from various out-patient radiology departments, the modality and quality of images was very heterogeneous, making a sound analysis not feasible. Due to the usually round or elliptic growth of BM we therefore calculated the volume based on the diameters.
- Did you account for leptomeningeal recurrence or progression?
Yes, we have included these patients in the in-brain control group and have counted them (very few) as in-brain recurrence.
- Were you using RANO or similar criteria to define progression?
As opposed to glioma, RANO criteria for BM are not regularly used in our department. Furthermore, they are difficult to apply in a retrospective study. In cases where it was difficult to distinguish between local recurrence, radiation necrosis and pseudoprogression under immune- or targeted therapy we used an additional FET-PET. We added this to the manuscript. (l. 143-146)
- You seem to indicate 29 patients died within 30d and could not be included? this is very high isn't it, like 10%? please clarify. Is it due to high rate of complications or poor performance status?
Yes, the mortality is in fact high. However, a minority were caused by surgical complications, but were rather due to the general condition of these patients with rapid systemic progression or severe comorbidity. We clarified this in the manuscript. (l. 165-167; 330-333)
- Can you rate eloquence in some way e.g. Sawaya grading system, left hemisphere vs right hemisphere, clarify what you mean by this - is it just speech, or motor areas or what
We did use the definition previously applied by Chang et al. adapted from low-grade gliomas to indicate a central region as well as speech and visual cortex as eloquent areas. (l. 126-128)
- Clarify lung subtype - assume NSCLC.
The lung cancers were exclusively NSCLC, we added this to the manuscript. (l. 171; Table 1)
- RPA group is very crude, even the GPA now is primary disease specific now, any attempt to use other scores like this?
We agree that a disease specific score would be preferable. But the heterogeneity of our cohort with respect to the primary tumours makes this impossible for this study, even more so since the implementation of molecular alterations would result in numerous small subgroups without any benefit for the key question of the paper. Therefore, we used the RPA classification since it has been well established and validated multiple times.
- The major factor here is the surgical technique - are you leaving residual because it is unsafe or because you are having variation in the surgery. SO much more detail please on the surgery - how many surgeons in this series? What standard techniques e.g. image guidance? ultrasound? What novel techniques like 5-ALA. Any adjuncts like motor mapping, awake or iMRI?
A very justified point – especially from a neurosurgeon! We added this information to the manuscript. As the use of 5-ALA is restricted to malignant glioma and would be off-label for brain metastases we did not use it. (l 111-115)
- It may be that your reasons for having great outcomes despite subtotal resection is just that you are doing something different with radiation - so you MUST clarify local the doses, machine (LINAC?), type of radiation - how many fractions, what do you do about dura, the margin, the cavity, when do you irradiate after surgery, what scans do you use to plan etc etc
We know that fractionated local radiotherapy concepts are not widespread yet. We have added substantial information regarding applied radiotherapy and included an experienced radio-oncologist (M. Kocher) as an author. (l 213-227)
- What is the role of immunotherapy in this series
A very complex issue and a very justified question!!! However, given the uncertainty of immune-checkpoint expression in BM and the heterogeneous effects reported in literature, a precise statement is not possible. Moreover, as the major cause of death was non-neurologic, the specific treatment effects of ICI on BM may not be deduced from these data. We addressed this point in the revised manuscript. (l 380-388)
- Table 3 is poorly displayed - do you mean this is univariate analysis of each factor or multivariate? Please state the time to LR and death in months for each level of the factors assessed e.g. >65y and <65 y old, GTR and STR then then univariate test statistic, then the multivariate if it was taken forward to multivariate analysis.
In fact, the table was mixed up by formatting making it absolutely incomprehensible. We adjusted this in the revised version! (Table 3)
- Only 60% of patients have died?? Line 208. This introduces significant lead time bias. Why are you reporting this series so early? Perhaps you should not include the more recent patients e.g. last couple of years. This could be a major issue.
The study was planned for summer 2020. Therefore, we have updated the follow-ups with more patients being dead at this time. However a high percentage of patients are still alive with a high fraction of long-time survivors, which is probably attributed to effective systemic treatment. We addressed this in the manuscript. Since one main question of this study is the local control rate, we are able to answer this after a radiological follow-up of one year (which already surpasses many survival rates in literature). (l 380-388, Table 3, Figure 2)
- Line 217 - please be much less vague and much more specific in your language, X increases Y, not "influenced" e.g. greater extent of resection was not associated with prolonged overall survival in this series (test statistic = X, p value = Y).
As written, influence is the intended meaning, as in Fig. 1; i.e. "extent" neither means greater or lesser, it is just a variable, so cannot be used here with increase (or prolonging OS). However, we added a clarifying statement as suggested. (l. 260-262)
Round 2
Reviewer 1 Report
One comment:
L 55 "Early studies demonstrated a significantly improved overall survival (OS) for patients with BMs receiving surgical resection compared to whole brain radiation therapy 55 (WBRT) [5,6]."
5. Vecht, C.J.; Haaxma-Reiche, H.; Noordijk, E.M.; Padberg, G.W.; Voormolen, J.H.C.; Hoekstra, F.H.; Tans, J.Th.J.; Lambooij, N.; 386 Metsaars, J.A.L.; Wattendorff, A.R.; et al. Treatment of Single Brain Metastasis: Radiotherapy Alone or Combined with Neuro-387 surgery. Annals of Neurology 1993, 33, 583–590, doi:10.1002/ana.410330605. 388
"The effect of neurosurgical excision plus radiotherapy was compared with radiotherapy alone in a prospectively randomized trial with 63 evaluable patients with systemic cancer and a radiological diagnosis of single brain metastasis. "
"We coclude that patients with single brain metastasis and stable extracranial tumor activity should be treated with surgical excision and radiotherapy. For patients with progressive extracranial disease during the previous 3 months, radiotherapy alone appears to be sufficient. After treatment of single brain metastasis, parients remain functionally independent until a few months before death."
6. Patchell, R.A.; Tibbs, P.A.; Walsh, J.W.; Dempsey, R.J.; Maruyama, Y.; Kryscio, R.J.; Markesbery, W.R.; Macdonald, J.S.; Young, 389 B. A Randomized Trial of Surgery in the Treatment of Single Metastases to the Brain. New England Journal of Medicine 1990, 322, 390 494–500, doi:10.1056/NEJM199002223220802.
"we randomly assigned patients with a single brain metastasis to either surgical removal of the brain tumor followed by radiotherapy (surgical group) or needle biopsy and radiotherapy (radiation group)."
"We conclude that patients with cancer and a single metastasis to the brain who receive treatment with surgical resection plus radiotherapy live longer, have fewer recurrences of cancer in the brain, and have a better quality of life than similar patients treated with radiotherapy alone. (N Engl J Med 1990; 322:494–500.)"
Therefore:
"receiving surgical resection compared to whole brain radiation therapy " is a misleading statement suggesting sited trials compared surgery alone to radiotherapy (RT) alone. Both references compared surgery+RT to RT. Please correct this.
Otherwise, the manuscript is much improved.
Author Response
# Reviewer 1
Thank you again for your time and effort!
L 55 "Early studies demonstrated a significantly improved overall survival (OS) for patients with BMs receiving surgical resection compared to whole brain radiation therapy 55 (WBRT) [5,6]."
- Vecht, C.J.; Haaxma-Reiche, H.; Noordijk, E.M.; Padberg, G.W.; Voormolen, J.H.C.; Hoekstra, F.H.; Tans, J.Th.J.; Lambooij, N.; 386 Metsaars, J.A.L.; Wattendorff, A.R.; et al. Treatment of Single Brain Metastasis: Radiotherapy Alone or Combined with Neuro-387 surgery. Annals of Neurology 1993, 33, 583–590, doi:10.1002/ana.410330605. 388
"The effect of neurosurgical excision plus radiotherapy was compared with radiotherapy alone in a prospectively randomized trial with 63 evaluable patients with systemic cancer and a radiological diagnosis of single brain metastasis. "
"We coclude that patients with single brain metastasis and stable extracranial tumor activity should be treated with surgical excision and radiotherapy. For patients with progressive extracranial disease during the previous 3 months, radiotherapy alone appears to be sufficient. After treatment of single brain metastasis, parients remain functionally independent until a few months before death."
- Patchell, R.A.; Tibbs, P.A.; Walsh, J.W.; Dempsey, R.J.; Maruyama, Y.; Kryscio, R.J.; Markesbery, W.R.; Macdonald, J.S.; Young, 389 B. A Randomized Trial of Surgery in the Treatment of Single Metastases to the Brain. New England Journal of Medicine 1990, 322, 390 494–500, doi:10.1056/NEJM199002223220802.
"we randomly assigned patients with a single brain metastasis to either surgical removal of the brain tumor followed by radiotherapy (surgical group) or needle biopsy and radiotherapy (radiation group)."
"We conclude that patients with cancer and a single metastasis to the brain who receive treatment with surgical resection plus radiotherapy live longer, have fewer recurrences of cancer in the brain, and have a better quality of life than similar patients treated with radiotherapy alone. (N Engl J Med 1990; 322:494–500.)"
Therefore:
"receiving surgical resection compared to whole brain radiation therapy " is a misleading statement suggesting sited trials compared surgery alone to radiotherapy (RT) alone. Both references compared surgery+RT to RT. Please correct this.
Otherwise, the manuscript is much improved.
Thank you very much for this intent comment, we incorporated the suggested, more precise statement.
Reviewer 3 Report
This is basically there now, just need to clarify if the different types of radiation e.g. WBRT vs. fractionated, were compared for their their local control rates in case this confounds the results on GTR/STR. thanks
Author Response
# Reviewer 3
Thank you again for your time and effort!
This is basically there now, just need to clarify if the different types of radiation e.g. WBRT vs. fractionated, were compared for their their local control rates in case this confounds the results on GTR/STR. thanks.
We have added this information to the table and text. (table 1, table 3, l. 222-223 and l.229-230).